# Identification of transformer overload and new energy planning for enterprises based on load forecasting

Longjin Lv[1], Yuxian Han[2]*

**1** School of Finance and Information, Ningbo University of Finance and Economics, Ningbo, China, **2** School of Automation, Central South University, Changsha, China

* yuxianhanchangsha@163.com

## Abstract

The new energy system constructed by energy storage and photovoltaic power generation system can effectively solve the problem of transformer overload operation in some enterprises. It can not only reduce the cost of electricity, but also realize low-carbon emission reduction. However, due to its low return on investment, the willingness of enterprises to install new energy is not high. In this paper, we first establish a load forecasting model to users whose transformers are overloaded or about to be overloaded, which are potential customers with new energy installation needs. Then, Optimal configuration models of PV and energy storage systems based on nonlinear programming are developed for these potential customers. The optimal installed capacity of the PV energy storage and the optimal charging and discharging strategy for the energy storage system can be obtained. This optimization strategy ensures that the electricity consumption of the enterprise does not exceed the rated capacity, and effectively reduces the enterprise's basic tariff and electricity price to achieve cost reduction and efficiency. Finally, taking a building materials production factory as an example, we obtain the optimal plan for the new energy capacity, as well as the economic benefits of the plan and the specific strategy of energy storage charging and discharging for this factory.

**Data Availability Statement:** All relevant data are within the manuscript and its Supporting Information files.

**Funding:** This work is supposed by the Major Humanities and Social Sciences Research Projects

## 1 Introduction

With the continuous growth of global energy demand and the profound change of energy structure, the stable operation of power system and the sustainable development of enterprises are facing unprecedented challenges. Power load forecasting, as a key link in power system planning and operation, is of great significance to guarantee the stability and reliability of power supply. At the same time, with the rapid development and popularization of new energy technology, enterprise new energy planning has gradually become an important way to promote the transformation and upgrading of enterprises and realize green development. In the power system, the transformer as the core equipment for power transmission and distribution, its operation status is directly related to the safe and stable operation of the whole system.

in Zhejiang higher education institutions (2023QN024), Zhejiang Provincial Natural Science Foundation of China (ZCLTGS24D0101). The funders had no role in study design, data collection and analysis, decision to publish, or preparation of the manuscript.

**Competing interests:** The authors have declared that no competing interests exist.

However, in the actual operation process, due to the volatility and uncertainty of the power load, the transformer often faces the risk of overload. Overloading not only leads to equipment damage and safety accidents, but also affects the overall operational efficiency of the power system. Therefore, transformer overload identification based on load prediction has become one of the current research hotspots in the field of power systems.

Load prediction, as a fundamental tool in power system analysis, plays a pivotal role in transformer overload identification and renewable energy planning. By accurately forecasting the future load demand, load prediction enables enterprises to anticipate potential transformer overloads and take proactive measures to avoid them. Furthermore, it provides valuable insights for the optimal planning and integration of renewable energy sources into the enterprise energy system, ensuring a reliable and sustainable power supply.

The grid load forecasting is important for the daily operation and maintenance of power grids, and there are many linear regression methods based on time series. The widely used ARIMA model and other technologies were organically combined to improve the prediction accuracy [1–4]. These methods all rely on past data to forecast, with simple structures, without considering the correlation between data, and only treating historical and predicted data as simple mathematical relationships, with low forecasting accuracy. Currently, machine learning and deep learning methods are also used for load prediction, such as XGBoost [5, 6], LSTM [7–9], neural networks [10–12], and the Bayesian model [13].The authors use improved robust estimation and fuzzy linear regression models to reduce prediction errors and improve robustness [14]; The authors firstly analyze the data by clustering and then build a prediction model for each category, which effectively improves the prediction accuracy [15, 16]. Although the above machine learning methods are effective in processing nonlinear data, they have weak feature extraction capability and low accuracy in forecasting highly random data.

On the other hand, with the expansion of production, the enterprise power consumption gradually rise, which led to the original installation of the transformer capacity is gradually insufficient, thus appearing overload phenomenon. The application of photovoltaic energy storage can prevent the distribution transformer large over current burnout and load switch frequent tripping problems, can completely solve the distribution transformer full overload problem, to ensure the safe and reliable operation of the enterprise. How to optimize the configuration of new energy capacity makes the enterprise economic benefits to reach the optimal is the current hot topic.

The electricity tariff paid by industrial large users not only includes the electricity tariff charged according to the actual electricity consumption, but also includes the basic electricity tariff related to the maximum demand of users. Therefore, for industrial enterprises, precise management of load electricity demand can reduce electricity demand and basic electricity costs. Under the regulation of time of use electricity prices, by controlling industrial loads, the fluctuation of loads can be reduced, resulting in a decrease in the electricity consumption of industrial enterprises and ultimately improving the overall economic benefits of the enterprise.

Nottrott et al. conducted scheduling modeling for energy storage to achieve peak net load management and minimize demand for photovoltaic energy storage systems. Research has shown that the proposed optimization strategy can significantly improve the net present value of photovoltaic systems [17]. Gupta et al. proposed a method to derive cost-optimal plans for countrywide deployment of PV generation and energy storage systems considering the MV power distribution infrastructure's technical limitations [18]. Based on the analysis of the uncertainty of renewable energy output and load demand, Elgamal et al. proposed an independent microgrid next day optimization scheduling scheme [19]. Ramírez et al. considered different scheduling scenarios and proposed a joint scheduling strategy for optimizing the

operation of microgrids, providing reference for the operation of island microgrids [20]. Taghikani studies the optimization scheduling problem of renewable energy microgrids and energy storage systems, introducing virtual power plants for load scheduling control and energy management [21].

The charging and discharging control optimization algorithm of battery energy storage system can reduce the peak-to-valley difference between the grid and the load [22, 23], and effectively solve the grid energy balance problem caused by the dual fluctuation of the new energy and the load. The charging and discharging control of battery energy storage systems [24–26] is constrained by the operating states, load characteristics, reliability indexes, and stability performance of new energy sources, traditional power sources, and power grids. An optimal scheduling model for PV-ESS comprehensively considering factors in terms of energy cost and charging/discharging constraints of the PV-ESS is proposed [27], Reference [28] introduces an optimization model for community energy markets that incorporates BESS to increase community income. The model assesses BESS degradation, explores different arrangements, and evaluates feasibility based on social welfare and fairness indicators, providing valuable insights into the impact of BESS within energy communities. In addition, reference [29] emphasizes the positive impact of integrating electricity storage systems into electrical networks, enhancing the stability of primary generators and improving overall system safety and reliability.

The integrated use of photovoltaic and energy storage systems is of great significance to manufacturing enterprises. First, through photovoltaic power generation to reduce the power grid load, reduce electricity costs, while reducing carbon emissions; Secondly, the energy storage system "low storage and high discharge" is used to effectively reduce the cost of electricity for enterprises. In addition, there is another important significance, which can effectively solve the problem of over-capacity electricity consumption in enterprises. Overcapacity electricity is power users exceed the contracted capacity of electricity, it will not only lead to transformers and other electrical equipment overload burn, cause fire, but also cause public power supply line overload, endangering the safety of the public grid. The survey found that some enterprises only have short-term overload in the case of equipment start-up and peak production. In order to meet the demand for safe electricity during peak load, enterprises have to carry out transformer expansion, which not only takes a long process but also costs a lot. According to conservative estimates, the expansion of 100kW requires a cost of about 200,000, and the expansion of the transformer will also bring about the rise of the basic electricity bill. The photovoltaic system can reduce the power grid load during the day, and the "peaking and valley filling" function of the energy storage system can effectively reduce the power grid load at other times, so the optical energy storage system can effectively avoid excess capacity power.

This paper aims to combine the load forecasting technology to accurately identify the overload state of the transformer and provide scientific basis for the new energy planning of the enterprise. Through load forecasting, the trend of power demand can be understood in advance, providing strong support for transformer operation scheduling and overload warning. At the same time, based on the results of load forecasting, enterprises can more accurately assess the demand and potential of new energy and formulate more reasonable new energy planning. This paper is arranged as follows: First of all, on the premise of knowing the current power load and rated capacity of the transformer of the enterprise, this paper uses big data technology to find out the specialized users whose load is expected to be near overload, and carries out accurate marketing of new energy for such enterprises. Then, combining enterprise load characteristics, ladder electricity price and energy storage cost, an optimal economic cost photovoltaic energy storage configuration model is established, and the optimal energy storage charging and discharging strategy is given. Finally, the economic benefits of new energy

governance are calculated. The research results can not only effectively avoid the excessive capacity of enterprise electricity consumption, but also realize the double reduction of electricity and basic electricity costs, which can significantly improve the economic benefits of industrial enterprise users.

Overall, the main contributions of this paper can be summarized as follows: 1)A load forecasting model based on Gaussian CNN-GRU is proposed to identify potential users for new energy installation, making the new energy configuration model more targeted. 2) An optimal configuration model for photovoltaic energy storage is established for users whose transformer capacity is already overloaded or about to overload. Compared with traditional models, the objective function considers the impact of the cost of transformer capacity upgrade and basic electricity price increase.3) A dual objective optimization model is proposed to minimize the payback period and investment cost of photovoltaic energy storage. The reason is that the current return cycle of energy storage is too long. Therefore, while overcoming the overload problem, the initial investment cost of energy storage should be minimized as much as possible.

## 2 Potential customer identification

### 2.1 Model construction

Whether an industrial enterprise can save electricity costs by installing new energy sources such as photovoltaic energy storage depends on many factors, the most important of which is its daily electricity load. Therefore, by forecasting the power load of industrial enterprises, this section excavates potential customers suitable for installing photovoltaic energy storage systems.

From the historical data, the power load characteristics show a gradual upward trend in a certain period. However, the power load is affected by many factors, such as holidays, seasons and other factors. It has its own unique pattern of change, the traditional time series analysis simulation is difficult to apply. In this section, the Gaussian-CNN-GRU prediction model is used to forecast the power load. This model is a neural network model combining feed forward neural network, convolutional neural network and gated cycle unit, and has improved the output layer and training method compared with the traditional load forecasting model. The network structure of the prediction model is shown in Fig 1.

The network input data is multi-characteristic time series data, including load variable, time variable, climate variable, economic variable and power pricing. The Gaussian CNN-GRU network consists of one input layer, three hidden layers, and one Gaussian output layer. Among them, the first hidden layer is a one-dimensional convolutional neural network, the second hidden layer is a gated cyclic unit, and the third hidden layer is a fully connected layer. The Gaussian output layer consists of two neurons that output two parameters, namely the predicted load value and the variance representing the uncertainty of the load prediction.

The first layer is the hidden layer (CNN layer), which is used to mine potential relationships between load data and variables such as climate and time, and extract high-dimensional features. The convolutional layer maps the enterprise power load information into $U$ time series through $U$ convolution cores. The time series length is shortened to $W$ by a step size greater than 1. Finally, the $U \times W$ matrix is output to the GRU layer.

The second layer is the hidden layer (GRU layer) for extracting timing characteristics. The power load information matrix is calculated step by step, and the $U \times 1$ format enterprise power load information vector is output from the last step neuron to the fully connected layer.

The calculation formula of the fully connected layer and the output layer is shown as follows:

$$\begin{bmatrix} \mu \\ \sigma \end{bmatrix} = W_{2 \times V} \cdot \sigma(W_{V \times U} \cdot h_{U \times 1} + b_{V \times 1}) + b_{2 \times 1}$$

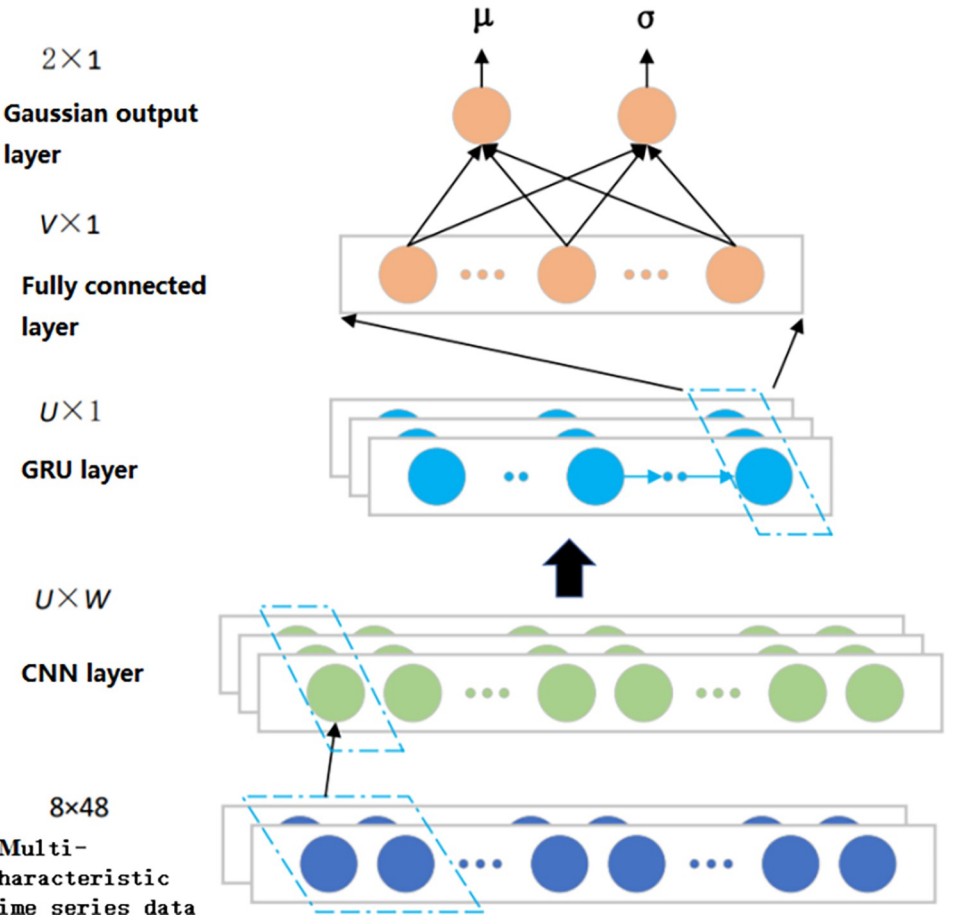

**Fig 1. Gaussian-CNN-GRU model structure.**

where, $\mu$ is the Load prediction value. $\sigma$ is the variance, denoting the uncertainty of load forecasting due to the randomness of enterprise power load. $h_{U\times1}$ represents the output vector of the GRU network to represent the electrical information. $W_{V\times U}$ and $b_{V\times1}$ are the weight matrix and bias vector of the fully connected layer, respectively. $W_{2\times V}$ and $b_{2\times1}$ are the weight matrix and bias of the output layer, respectively.

## 2.2 Result prediction

Take a building materials production enterprise as an example. The data of electricity load, economy, weather and electricity price of the company from January to June 2023 were collected, and 96 electricity load data from January to June were used as the original input data in a 15-minute period. A new stationary series was obtained after smooth processing, and the Gaussian-CNN-GRU electricity load prediction model was established. Finally, the electricity load forecast results for 2023 are obtained, and the comparison between the predicted value and the actual value is shown in Fig 2. Using the actual electricity load data from June to December 2023 as verification data, the error is shown in Fig 3. Through error comparison analysis, the Gaussian-CNN-GRU model is better than the ARIMA model in power load prediction.

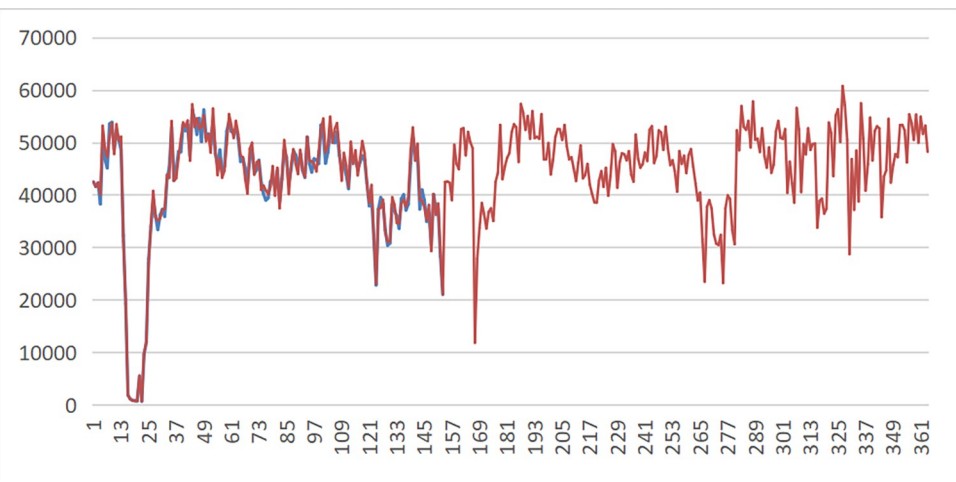

**Fig 2. Results of electricity load forecasting.**

In order to prove the effectiveness of the model, we calculated the root-mean-squared error and R squared. The calculation formulas are as follows:

$$RMSE = \sqrt{\frac{1}{n}\sum_{i=1}^{n}\left(\hat{y}_i - y_i\right)^2} = 1.1423 \tag{1}$$

$$R^2 = 1 - \frac{\sum_{i=1}^{n}\left(\hat{y}_i - y_i\right)^2}{\sum_{i=1}^{n}\left(y_i - \bar{y}_i\right)^2} = 99.38\% \tag{2}$$

where $y_i$, $\bar{y}_i$ and $\hat{y}_i$ represent the real value, the average value, and the predicted value of the model of the load at a certain time in the sequence, respectively. These indicators once again show that the prediction error of the model is small and the fitting degree is high. The model can effectively portray the cyclical, seasonal, and holiday influences of load data, effectively improving the forecasting accuracy.

Next, we use the prediction model to identify potential customers for identifying new energy installations based on the following: when the maximum value of the load forecast in the next three months $\geq$ 80% of the dedicated variable capacity in the marketing system. All

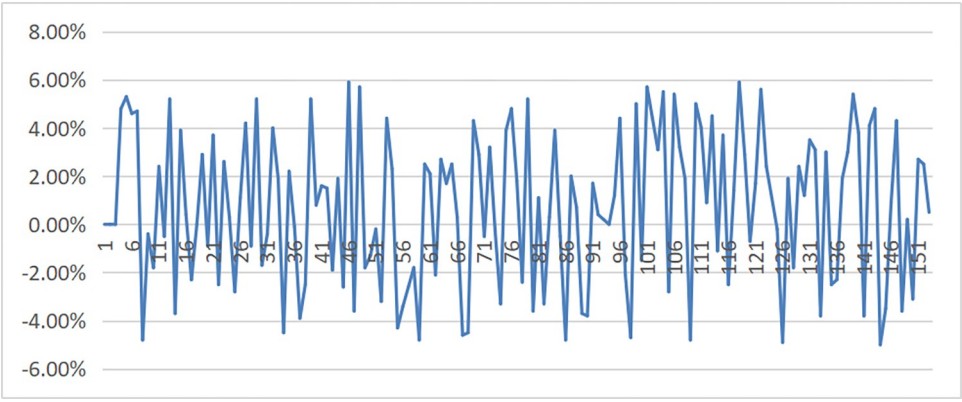

**Fig 3. Errors in Gaussian-CNN-GRU prediction models.**

dedicated variable customers above 300kva in a region are selected, totaling 820. Through the model, 89 of them are found to meet the identification requirements, and they are ranked according to their overload magnitude from largest to smallest.

## 3 Optimal configuration and operation strategy of the new energy system for users with transformer overload

### 3.1 Variable definition

Assume that the power generated by the photovoltaic system at time $t$ is $P_{PV}(t)$. Considering that the maximum power output by photovoltaic is about 80% of the transformer capacity, the first variable to be optimized is the maximum power output by photovoltaic $C_{PV}$. The other variables to be optimized are the energy storage system capacity $C_{ES}$, and the maximum charging and discharging power of the energy storage system $P_{ES}^{max}$.

The intermediate variables in the optimization model include the following variables: the power $P_{PV-PG}(t)$ provided by the photovoltaic system to the power grid, the power $P_{PV-PF}(t)$ provided by the photovoltaic system to the production factory, and the power $P_{PV-ES}(t)$ supplied by the photovoltaic system to the storage for charging the energy system, the power $P_{PG-ES}(t)$ for charging the energy storage system purchased from the power system, and the power $P_{PG-PF}(t)$ purchased from the power system to supply the production plant, and the power $P_{ES-PF}(t)$ discharged by the energy storage system to the production plant. These six variables are key technical indicators for maximizing the benefits of the configuration system and their relations can been found in Fig 4. In addition, suppose that the power load function of the production plant is $P_{PF}(t)$, and the output of photovoltaic power generation per unit installed capacity is $P_{PV}(t)$, and the capacity of the company's existing transformer is $C_{PG}$, and the pricing function of electricity charges in different periods is $COS_{PG}(t)$. These data can be regarded as known functions or known quantities in model analysis.

### 3.2 Optimal configuration of photovoltaic energy storage

The objective function and constraints of the optimization model will be established below.

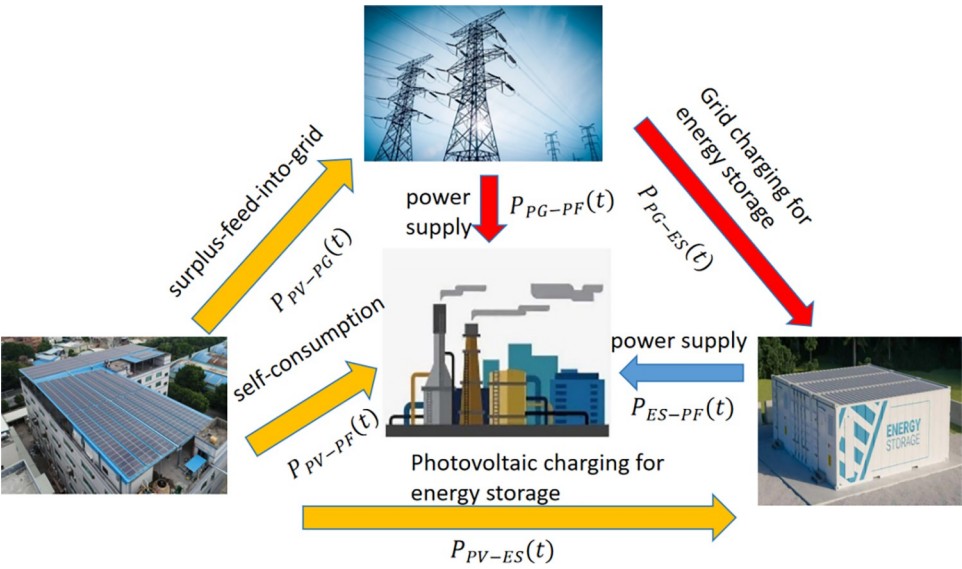

**Fig 4. The relationship between key variables in the optimal configuration model.**

**3.2.1 Objective function.** We select the minimization of PV energy storage payback period as the optimization objective. At the same time, because the application of energy storage is mainly to solve the problem of transformer capacity overloading, and the current energy storage payback period is long, so the initial investment cost of energy storage should be reduced as much as possible under the premise of ensuring that the electricity load of the enterprise power grid is within the safe power range. Therefore, the minimization of investment cost of energy storage is added to form a double objective function as follows

$$min\{TC/PRO\} \tag{3}$$

$$min\{COS_{ES}\} \tag{4}$$

Here *TC* is the new energy system construction cost, including the construction cost of the photovoltaic system $COS_{PV}$, and the construction cost of the energy storage system $COS_{ES}$. It should also be subtracted the transformer capacity upgrade cost $COS_{PG}$ and the government subsidy income $COS_{PG}$. So the *TC* is given as

$$TC = COS_{PV} + COS_{ES} - COS_{PG} - PRO_{GS} \tag{5}$$

Here $COS_{PV}$ is the construction cost of the photovoltaic system, given by $COS_{PV} = c_{PV} C_{PV}$, where $c_{PV}$ is the photovoltaic unit construction cost.

The $COS_{ES}$ is the energy storage construction cost, which is proportional to the installed capacity and charging and discharging power, given as

$$COS_{ES} = c_{ESC}*C_{ES} + c_{ESP}*P_{ES}^{max} \tag{6}$$

where $c_{ESC}$ is the construction cost of the installed capacity of the energy storage unit, and $c_{ESP}$ is the maximum charging capacity of the energy storage unit Discharge power construction cost.

The $COS_{PG}$ *is* the transformer capacity upgrade cost, given as $COS_{PG} = c_{PG} \Delta C_{PG}$, where $c_{PG}$ is the unit cost of the transformer upgrade, and $\Delta C_{PG}$ is the capacity size of the company's transformer that needs to be upgraded based on the company's electricity consumption forecast;

The $PRO_{GS}$ is the government subsidy income, given as
$PRO_{GS} = c_{PVS}* \int_{1}^{8760n_s} P_{PV}(t)dt + c_{ESS}*C_{ES}$, where $c_{PVS}$ represents the subsidy of photovoltaic unit power generation, $c_{ESS}$ represents the subsidy of unit energy storage installed capacity, $n_s$ represents the number of years of photovoltaic subsidy, and 8760 represents the number of hours in a year.

The annual net operating income PRO of the new energy system is given as

$$PRO = PRO_{PV-PF} + PRO_{PV-ES} + PRO_{PV-PG} + PRO_{ES} + PRO_{PG} \tag{7}$$

The electricity cost saved by the photovoltaic output to the production plant $PRO_{PV-PF}$, the energy storage system $PRO_{PV-ES}$, and the electricity fee income obtained by connecting the photovoltaic surplus to the grid $PRO_{PV-PG}$ are the product of the corresponding unit electricity price and electricity, that is

$$PRO_{PV-PF} = \int_{1}^{8760} COS_{EP}(t)*P_{PV-PF}(t)dt \tag{8}$$

$$PRO_{PV-ES} = \int_{1}^{8760} COS_{EP}(t)*P_{PV-ES}(t)dt \tag{9}$$

$$\text{PRO}_{\text{PV-PG}} = c_{\text{PVS}} * \int_{1}^{8760} P_{\text{PV-PG}}(t) dt \tag{10}$$

where $\text{COS}_{\text{EP}}(t)$ represents the power system time-of-use unit electricity price function, which is a piecewise function.

The electricity bill difference obtained from the energy storage system's "peak shaving and valley filling" $\text{PRO}_{\text{ES}}$ refers to the difference between the electricity bill during the charging and discharging of the energy storage system, given as

$$\text{PRO}_{\text{ES}} = \int_{1}^{8760} \text{COS}_{\text{EP}}(t) * P_{\text{ES-PF}}(t) dt - \int_{1}^{8760} \text{COS}_{\text{EP}}(t) * (P_{\text{PV-ES}}(t) + P_{\text{PG-ES}}(t)) dt \tag{11}$$

The $\text{PRO}_{\text{PG}}$ is the basic electricity savings from energy storage replacement transformer upgrades, which is proportional to the $\Delta C_{PG}$, given as $\text{PRO}_{\text{PG}} = c_{\text{bp}} \Delta C_{PG}$, where $c_{\text{bp}}$ is the unit cost of the basic electricity savings.

**3.2.2 Constraints condition.** *a) Linear inequality constraints.* When the installed photovoltaic capacity of the enterprise is, the photovoltaic power generation output at time t should satisfy the following two formulas:

$$P_{\text{PV}}(t) = C_{\text{PV}} * p_{\text{PV}}(t) \tag{12}$$

$$P_{\text{PV}}(t) = P_{\text{PV-PF}}(t) + P_{\text{PV-ES}}(t) + P_{\text{PV-PG}}(t) \tag{13}$$

In addition, according to the conditions of photovoltaic power generation, the maximum power of photovoltaic output is about 80% of the installed photovoltaic capacity: $\max\{P_{\text{PV}}(t)|0 \leq t \leq 24\} = C_{\text{PV}} * 80\%$. The power $P_{\text{PG}}(t)$ supplied by the power system should meet the following conditions:

$$P_{\text{PG}}(t) = P_{\text{PG-ES}}(t) + P_{\text{PG-PF}}(t) \tag{14}$$

$$P_{\text{PG}}(t) \leq C_{\text{PG}} * 80\% \tag{15}$$

The power load $P_{\text{PF}}(t)$ of the production plant is provided by three parts: the part provided by the photovoltaic system, the power purchased from the power system, and the power provided by the discharge of the energy storage system, i.e.,

$$P_{\text{PF}}(t) = P_{\text{PV-PF}}(t) + P_{\text{PG-PF}}(t) + P_{\text{ES-PF}}(t) \tag{16}$$

Here, the power load $P_{\text{PF}}(t)$ is the input variable of the model, provided by the short-term forecast results in the Sec.2. It should be noted that if the short-term forecast results are changed to long-term forecast results, the optimal configuration model is still feasible.

At any time, the charging and discharging power of the energy storage system must be within the rated range and cannot exceed the rated maximum charging power $P_{\text{ES}}^{\text{max}}$, so there are the following constraint inequalities:

$$P_{\text{PV-ES}}(t) + P_{\text{PG-ES}}(t) \leq P_{\text{ES}}^{\text{max}} \tag{17}$$

$$P_{\text{ES-PF}}(t) \leq P_{\text{ES}}^{\text{max}} \tag{18}$$

When the energy storage system is charging or discharging, the power $S_{ES}(t)$ of the energy storage system at time t should satisfy

$$S_{ES}(t) = S_{ES}(t-1) + k(P_{PG-ES}(t) + P_{PV-ES}(t)) - P_{ES-PF}(t)/d \qquad (19)$$

Let k and d denote the charging and discharging efficiency of the energy storage system respectively, and the values are usually k = 0.98 and d = 0.98. The real-time power of the energy storage system is always within the rated capacity of the system:

$$0 \le S_{ES}(t-1) + k(P_{PG-ES}(t) + P_{PV-ES}(t)) - P_{ES-PF}(t)/d \le C_{ES} \qquad (20)$$

*b) Nonlinear constraints.* Since the energy storage system cannot be operated for charging and discharging at the same time, nor can it be charged with PV and the grid at the same time, the following constraints must be satisfied:

$$P_{PG-ES}(t) * P_{ES-PF}(t) = 0 \qquad (21)$$

$$P_{PV-ES}(t) * P_{ES-PF}(t) = 0 \qquad (22)$$

$$P_{PG-ES}(t) * P_{PV-ES}(t) = 0 \qquad (23)$$

c) Upper and lower bound constraints.

$$0 \le P_{PV-PG}(t), P_{PV-PF}(t), P_{PG-ES}(t) \le P_{PV}(t); \qquad (24)$$

$$0 \le P_{PG-ES}(t), P_{PG-PF}(t) \le P_{PG}(t); \qquad (25)$$

$$0 \le P_{ES-PF}(t), P_{PV-ES}(t), P_{PG-ES}(t) \le P_{ES}^{max}; \qquad (26)$$

$$P_{ES}^{max}, C_{ES}, C_{PV} \ge 0. \qquad (27)$$

## 3.3 The idea of solving the optimization model

Considering that the photovoltaic output function and the enterprise power load function are all discrete values (usually collected every 15 minutes to obtain past values, and predict future values accordingly), and the time-of-use electricity price is also a segmented function, in the calculation, we discretize the theoretical continuous function. The process of solving this optimization model is shown in Fig 5. The known data that needs to be input include: time-of-use predicted values for photovoltaic power generation, time-of-use predicted values for enterprise electricity loads, three-stage (valley, peak, peak) electricity prices, photovoltaic grid-connected electricity prices, photovoltaic and energy storage construction costs, and government subsidies.

Since the model is a multi-objective optimization model and cannot be solved directly, we adopt a two-stage approach. In the first step, we ignore Objective 2 and solve the optimization model with only Objective 1:min{*TC/PRO*}, keeping the constraints consistent with the original model. We assume that the investment payback period obtained in this step is N years. In the second step, we use Objective 2 of min{*COS_ES*} as the single objective function, taking the objective value from the first step as a constraint,i.e., adding the following constraint to the original model's constraints:*TC/PRO≤N*. The solution obtained in the second step will be the final solution of the model. In the process of solving the optimization model using the above two-step method, each step's optimization was performed using the optimization function 'fmincon' in MATLAB (as shown in Fig 5). The final optimized solution includes the

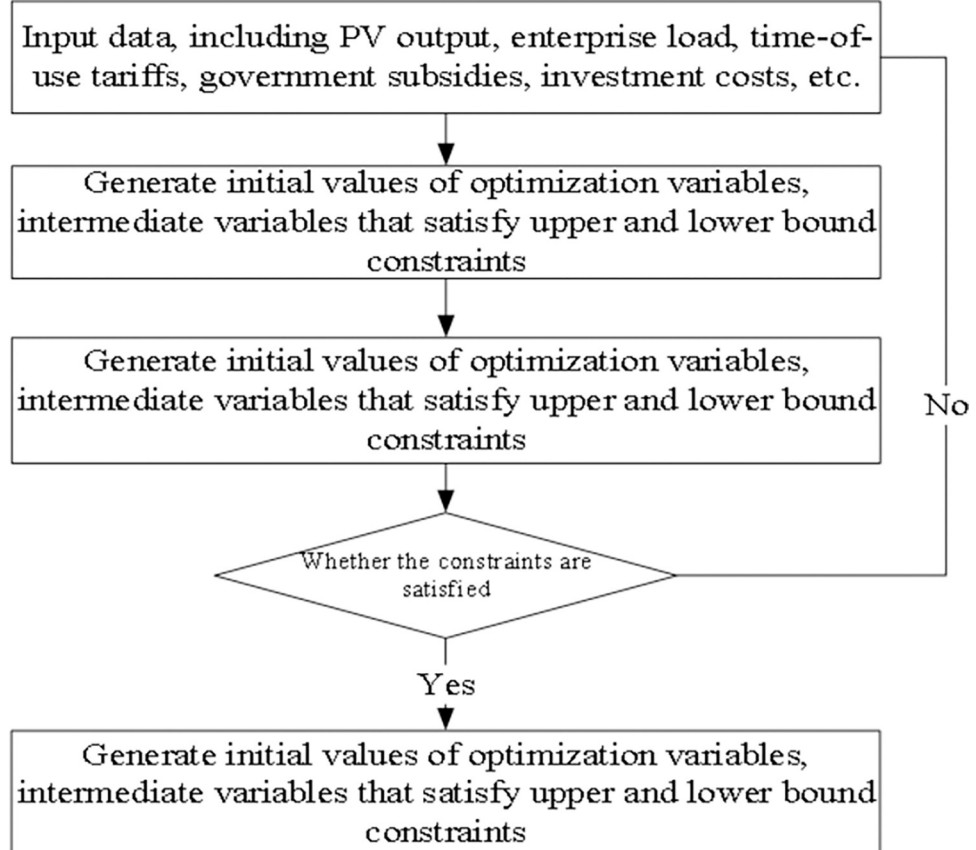

**Fig 5. The process of solving this optimization model.**

photovoltaic installed capacity, energy storage system capacity and power, power distribution among photovoltaics, energy storage, production plants, and power systems, as well as the business payback period and annual income.

## 3.4 Example analysis

In this section, A Building Materials production factory (Factory A) is selected for case analysis, which has not yet deployed photovoltaic and energy storage systems. We will give a detailed plan for the factory's photovoltaic and energy storage construction and configuration, as well as analyze the economic benefits after adopting the system.

**3.4.1 Optimal photovoltaic storage new energy system configuration.** Factory A's transformer capacity is 1000kW. According to the relevant regulations on power system power consumption, the power load of the factory's power system needs to be within 80% of the rated capacity of the transformer (that is, 800kW). The typical daily power load in June of the next year is predicted by the model as shown in Fig 6, the distribution of its daily electricity consumption in June is shown in Fig 7. It is easy to see that according to the forecast results, the factory's electricity load exceeds the safe power of the transformer (800kW) in many periods, so the factory should either replaces a larger-capacity transformer (expected to need more than 1100kW) or adopts a new energy system with photovoltaic energy storage.

Through the model calculation, the optimal configuration of the new energy system for this factory is given as follows: Current transformer rated capacity: 1000KW (safety power 800

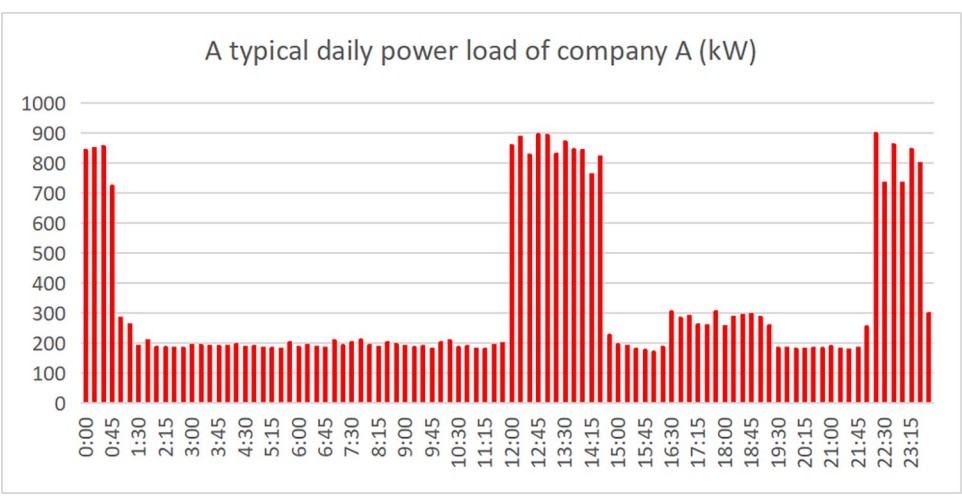

**Fig 6. Typical daily electricity load of HT Building Materials in June.**

KW); Photovoltaic capacity: 0.8930MW; Energy storage system capacity: 921kWh; Energy storage maximum charge and discharge power: 460kW.

Adopting the above-mentioned new energy system configuration of photovoltaic energy storage brings great economic benefits. The detailed indicators are presented in Table 1,and the subdivided income in June is shown in Table 2.

**3.4.2 The charging and discharging scheme of energy storage system for "Peak shaving and valley filling".** From Fig 8, we can see that after configuring the photovoltaic energy storage system, the peak load of factory A's power system has been effectively reduced, which indicates that the factory's overload issue has been resolved.

In Fig 9, we show the electricity load and photovoltaic output of enterprises on a typical day in June. The supply of the power system is shown in Fig 10. When the power load is negative, the remaining photovoltaic power is connected to the power grid. The peak value between 11–15 o'clock drops under the photovoltaic output, especially the peak value between 11–12 o'clock drops sharply. However, the peak value of the factory's 0–1 point and 22–24 point is relatively high, which exceeds the safe power (800kW) and needs to be "peak cut". In addition, the peak hours of 9–11 o'clock, 15–17 o'clock, 8–9 o'clock, 13–15 o'clock, and 17–22 o'clock can also reduce electricity charges through "high discharge". The specific energy storage charging and discharging scheme is shown in Fig 11, and the effect is shown in Fig 12.

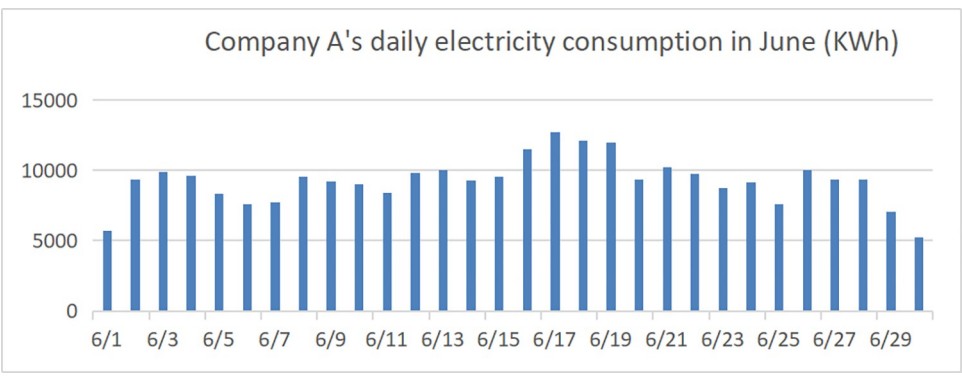

**Fig 7. Daily electricity consumption of HT building materials in June.**

**Table 1. Investment income of factory A's new energy system.**

| Construction cost TC ($10^4$ RMB) | Average annual income ($10^4$ RMB) | Payback period (years) | Average annual return on investment (10-year period) |
|---|---|---|---|
| 513.97 | 101.54 | 5.06 | 15.9% |

The power load of the enterprise first consumes photovoltaic power generation. When it is insufficient, the power grid is needed to supply power. It can be seen from Figs 10 and 11 that the power load exceeds the safe power of the transformer during the two time periods of 0–1 o'clock and 22–24 o'clock. Therefore, the energy storage discharges slightly in these two time periods for "peak shaving" to control the power supply power of the power grid within the safe power limit. The 9–11 and 15–17 o'clock are the peak electricity price periods, but since the electricity load of enterprises from 9–11 o'clock is less than that of photovoltaic power generation, energy storage power supply is not required, the energy storage is preferentially discharged during the peak period of 15–17 o'clock to reduce this time load. The remaining power of the energy storage system is used for the three peak electricity price periods of 8–9 o'clock, 13–14 o'clock, and 18–20 o'clock to reduce the power load of the power system; 21–24 o'clock, 7–8 o'clock and 11–13 o'clock is the valley price period, and the energy storage is charged as much as possible under the premise that the power load does not exceed the safe power of the transformer. It is worth noting that 11–13 points are in the valley price stage. However, since the on-grid price of surplus photovoltaic power (about 0.42 yuan/kWh) is higher than the valley price (about 0.26 yuan/kWh), the strategy of charging energy storage by the power system is adopted. In summary, from an intuitive point of view, the optimal charging and discharging strategy obtained by the mathematical optimization model is very reasonable.

We can use this method to analyze the economic benefits of new energy investment for all potential dedicated variable users, and gradually screen them from high to low according to the return on investment, which can greatly enhance the success rate of new energy promotion.

## 4 Conclusions

Facing the problems of low return on new energy investment, low willingness of enterprises to install new energy, and difficulty in promoting new energy, in this paper, we identify potential customers for new energy installation demand by building a forecast model of enterprise electricity load. By constructing a nonlinear mathematical optimization model to solve the optimal configuration of the photovoltaic system. In the case of known enterprise current time-sharing electricity load and the rated capacity of the transformer, by establishing an optimal configuration model, we find the optimal configuration scheme of enterprise photovoltaic energy storage and its revenue period, annual revenue and annual return on investment. Taking a building materials production factory as an example, the construction plan of photovoltaic energy storage new energy system of the factory is given, as well as the economic benefits of the plan and the specific strategy of energy storage charging and discharging are analyzed. The

**Table 2. Revenue of factory A's new energy system in June.**

| Arbitrage generated by "shaving peaks and filling valleys" | Income generated from photovoltaic surplus electricity connected to the grid | Income from self-consumption of photovoltaic power generation | Electricity benefit saved by energy storage "peak shaving" | Total |
|---|---|---|---|---|
| 28545 RMB | 6076 RMB | 47133 RMB | 11007 RMB | 92761 RMB |

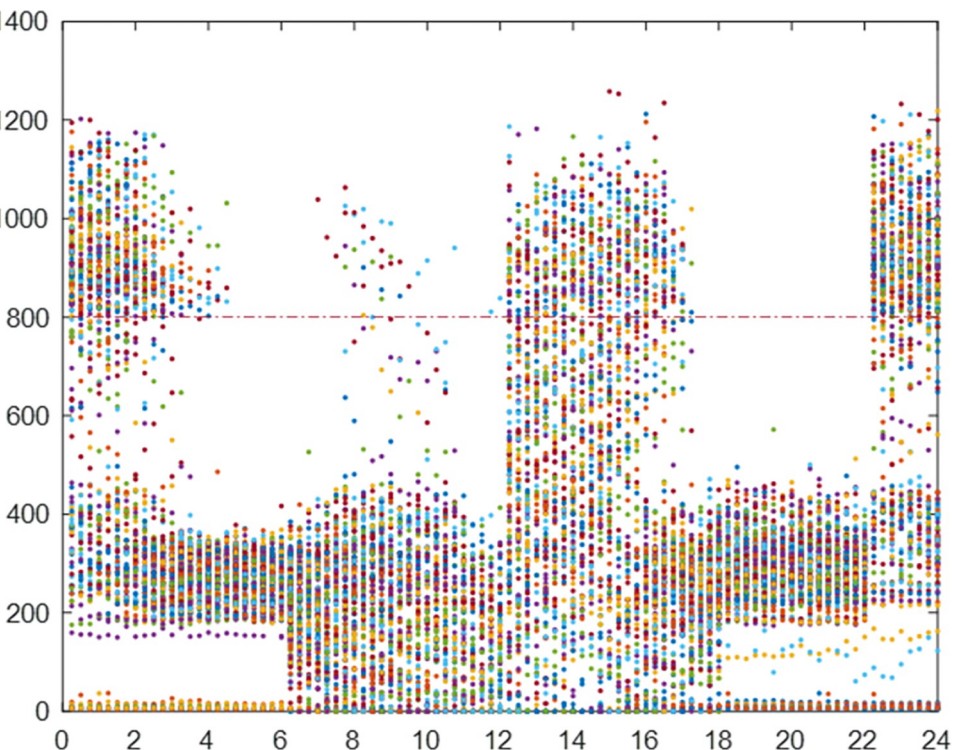

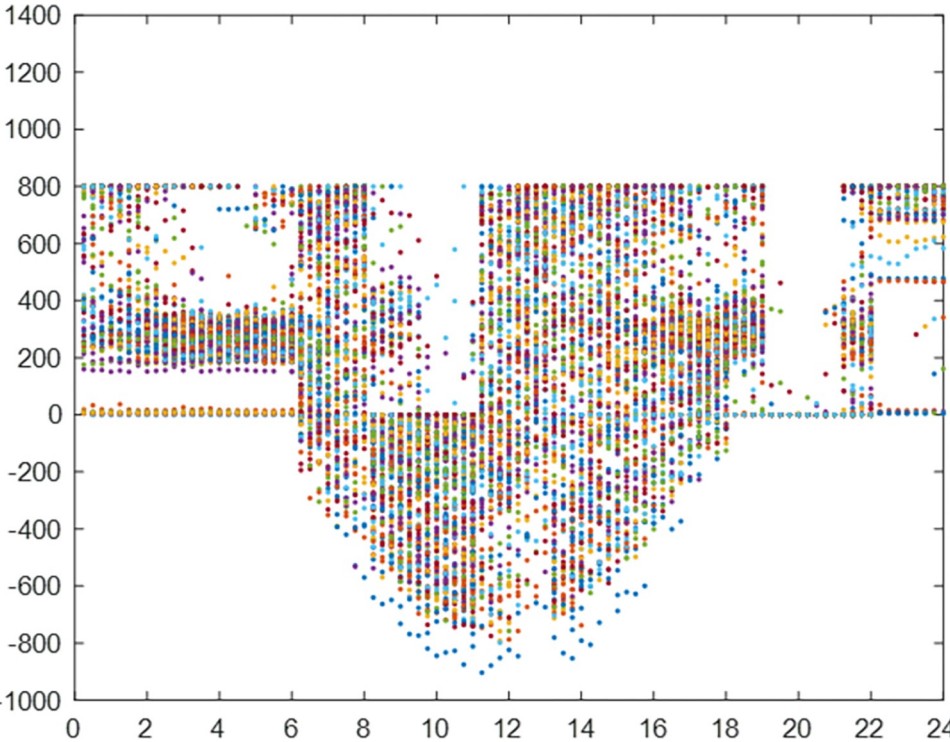

**Fig 8.** Power grid load of factory A before (a) and after (b) adopting the new energy system in June.

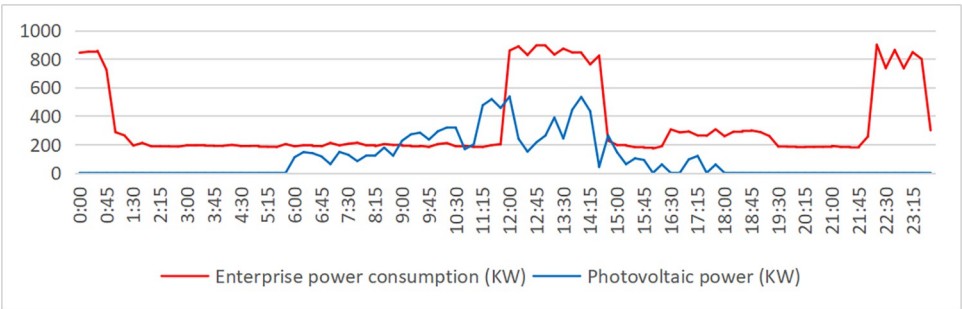

**Fig 9. Daily electricity consumption and photovoltaic power generation of factory A in June.**

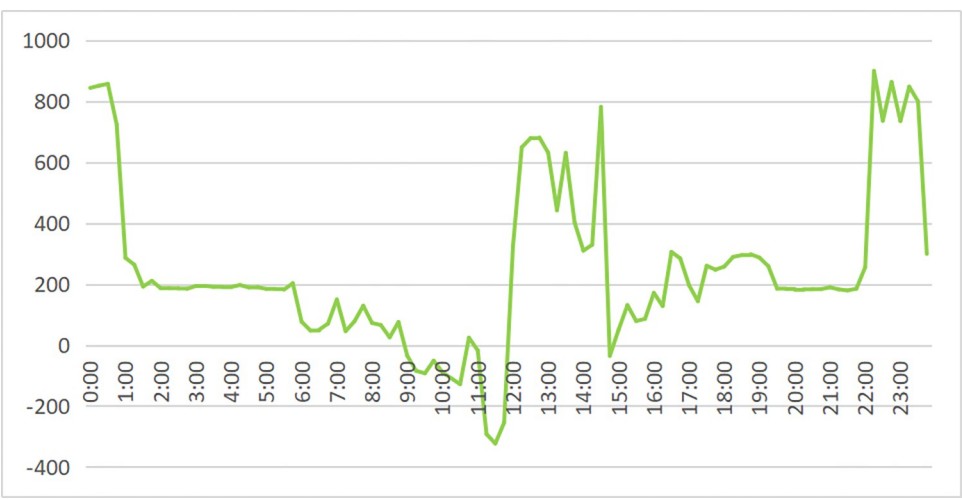

**Fig 10. Daily power supply of factory A in June with photovoltaic power generation enabled.**

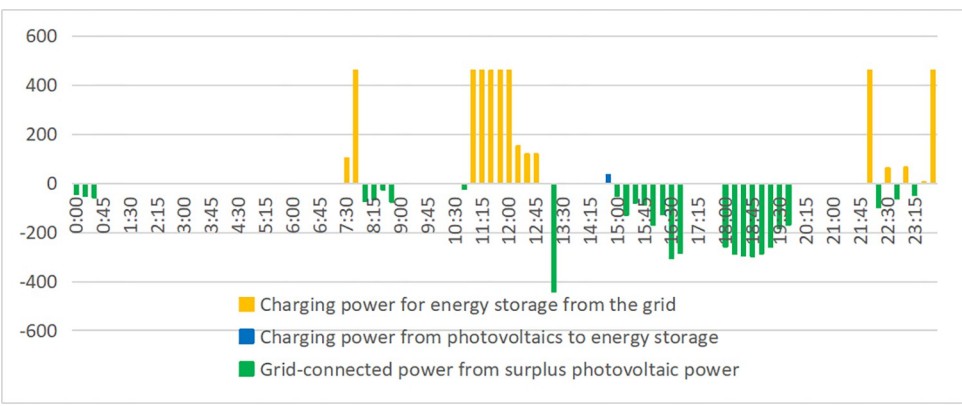

**Fig 11. Factory A's typical daily energy storage system charging and discharging strategy.**

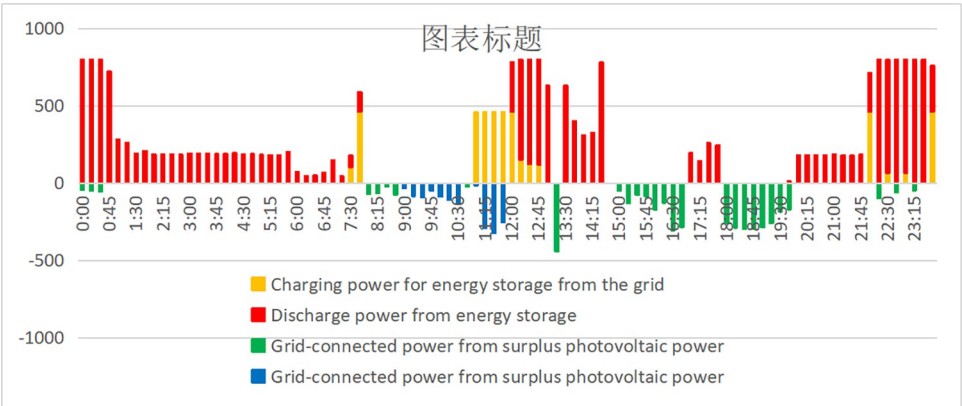

**Fig 12. Factory A's new energy system charging and discharging strategy on a typical day.**

research results of the paper help the power system enterprises to put forward targeted investment proposals for the enterprise photovoltaic energy storage new energy system, so as to effectively improve the power service to users.

## Supporting information

**S1 Data. The power load of factory A.**
(XLSX)

## Author Contributions

**Conceptualization:** Longjin Lv.

**Data curation:** Longjin Lv.

**Formal analysis:** Yuxian Han.

**Funding acquisition:** Longjin Lv.

**Investigation:** Longjin Lv, Yuxian Han.

**Methodology:** Longjin Lv.

**Resources:** Yuxian Han.

**Software:** Yuxian Han.

**Supervision:** Longjin Lv.

**Validation:** Longjin Lv, Yuxian Han.

**Visualization:** Yuxian Han.

**Writing – original draft:** Longjin Lv, Yuxian Han.

**Writing – review & editing:** Longjin Lv, Yuxian Han.

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
