## [Decision Letter · Decision Letter 0]

16 Jul 2024

PONE-D-24-16376Identification of transformer overload and new energy planning for enterprises based on load forecastingPLOS ONE

Dear Dr. Han,

Thank you for submitting your manuscript to PLOS ONE. After careful consideration, we feel that it has merit but does not fully meet PLOS ONE’s publication criteria as it currently stands. Therefore, we invite you to submit a revised version of the manuscript that addresses the points raised during the review process.

We look forward to receiving your revised manuscript.

Kind regards,

Ch. Rami Reddy, Post Doc

Academic Editor

PLOS ONE

“This work is supposed by the Major Humanities and Social Sciences Research Projects in Zhejiang higher education institutions(2023QN024), Zhejiang Provincial Natural Science Foundation of China (ZCLTGS24D0101).”

“This work is supposed by the Major Humanities and Social Sciences Research Projects in Zhejiang higher education institutions(2023QN024), Zhejiang Provincial Natural Science Foundation of China (ZCLTGS24D0101).”

“This work is supposed by the Major Humanities and Social Sciences Research Projects in Zhejiang higher education institutions(2023QN024), Zhejiang Provincial Natural Science Foundation of China (ZCLTGS24D0101).”

5. We note that your Data Availability Statement is currently as follows: [All relevant data are within the manuscript and its Supporting Information files.]

Reviewers' comments:

Reviewer's Responses to Questions

**Comments to the Author**

1. Is the manuscript technically sound, and do the data support the conclusions?

Reviewer #1: No

Reviewer #2: Yes

2. Has the statistical analysis been performed appropriately and rigorously? 

Reviewer #1: N/A

Reviewer #2: Yes

3. Have the authors made all data underlying the findings in their manuscript fully available?

Reviewer #1: No

Reviewer #2: No

4. Is the manuscript presented in an intelligible fashion and written in standard English?

Reviewer #1: No

Reviewer #2: Yes

5. Review Comments to the Author

Reviewer #1: 1. The abstract and the main body of this paper do not reflect the meaning of the title. It is more like an optimal configuration paper, rather than transformer overload identification paper. the authors should clearly check the whole paper.

2. Please do not lump the reference, but introduce the main contributions of each reference.

3. The research gap should be further concluded, and the main contributions should be re-organized as bullet-point style.

4. Considering the optimal configuration issue, the load forecasting results should also include the results for next 10-20 years. Usually, the planning years should be 15 or 20 years.

5. Section 2.2 should be included in the simulation section.

6. In section 3.2.1, there are two objectives, making the problem as a multi-objective optimization problem. Thus, the authors should discuss how to obtain the multi-objective optimization results. The “fmincon” function is not supposed to be the correct solution algorithm.

7. The integration of the forecasting results, especially in future years, into the optimization model should be further clarified.

8. Planning or configuration problem is usually a mix-integral problem, with the number of the equipment as the integers. The authors should discuss why the integer issue is not considered in this paper.

9. Direct planning or configuration results of the paper is missing.

10. The results should be compared with other existing methods to show the innovation.

11. The overload issue is not discussed in the case study.

12. The overall English expression should be improved.

Reviewer #2: The paper titled “Identification of transformer overload and new energy planning for enterprises based on load forecasting” is reviewed and I would like to appreciate the authors in presenting the work in a most significant way. But it requires the revisions such as

1. The authors mentioned a proposed overload prediction enterprise methodology for transformers by taking the energy storage system and PV system as examples. Draw and explain your key findings with a load flow diagram and load indication points.

2. The authors stated that your non-linear programming models detect the potential consumers who are under overload. How does this reduce the enterprise's basic tariff and electricity prices and improve efficiency? You did not mention the calculations and descriptions.

3. If a fault occurs in the systems, how does your methodology work? Is there any protection and identification system on the transformer end?

4. English grammar corrections are required, particularly on page 19, 1st paragraph. Check the grammar and technical corrections.

5. Apart from the above-mentioned points, the remaining article is technically sound and explained in a good way.

6. PLOS authors have the option to publish the peer review history of their article (what does this mean?). If published, this will include your full peer review and any attached files.

Reviewer #1: No

Reviewer #2: No

---

## [Author Response · Author response to Decision Letter 0]

23 Aug 2024

Response to Reviewer #1

1. Reviewer’s comments: The abstract and the main body of this paper do not reflect the meaning of the title. It is more like an optimal configuration paper, rather than transformer overload identification paper. the authors should clearly check the whole paper.

Response: In this paper, we first develop a load forecasting model to identify users with overload potential. In this category, in order to meet the demand for safe electricity during peak load periods, companies must undergo transformer expansion, which not only takes a long time, but is also very costly. It is conservatively estimated that a 100kW expansion will cost about $200,000, and the transformer expansion will also result in an increase in the basic electricity bill. We take such enterprises as potential customers for PV energy storage installation. PV energy storage system can not only effectively avoid the enterprise's excessive electricity capacity without the need to expand the transformer, but also can realise the double reduction of the cost of electricity and basic electricity. Basically, we establish the optimal economic cost PV energy storage allocation model by combining the enterprise load characteristics, ladder tariff and energy storage cost, and give the optimal energy storage charging and discharging strategy. The whole paper is centred on the two aspects of the title. This paper proposes a PV energy storage optimal configuration model for companies with transformers that are already overloaded or about to be overloaded, which is more economically efficient than the unconstrained PV energy storage optimal allocation model. We made changes in the abstract and Introduction to bridge the front and back.

2.Reviewer’s comments: Please do not lump the reference, but introduce the main contributions of each reference.

Response: We have reorganized the contributions of the references, for details, please refer to Sec.1.

3.Reviewer’s comments: The research gap should be further concluded, and the main contributions should be re-organized as bullet-point style.

Response: We have summarized the main contributions of this paper, which can be found at the end of Sec.1.

4. Reviewer’s comments: Considering the optimal configuration issue, the load forecasting results should also include the results for next 10-20 years. Usually, the planning years should be 15 or 20 years.

Response:This article aims to explore customers whose transformer capacity is about to overload through load forecasting, and identify these customers as potential high-quality customers for new energy installation. A new energy optimal configuration model is proposed for these customers. We selected 820 enterprises in a certain region with transformer capacities greater than 300KVA as research objects, and screened out 89 potential users. Under the constraint of not overloading the load after the installation of new energy, we established the optimal configuration model for new energy. In the paper, we only studied short-term load expectations, mainly because: 1) we aim to identify users who are about to overload in the near future. 2) Most of the 820 selected companies only have data every 15 minutes for the past 3 years, which makes it impossible for us to make accurate long-term predictions. Long term forecasting models can be our future research content, but this requires to introduce macroeconomic and other data. After getting the long term forecasting results，we only need to change the short-term prediction results in the input variables of the optimal configuration model to long-term predictions, then we can obtain the long-term planning results for the enterprises. We add this explanation below Eq.(16) in Page 8. 

5.Reviewer’s comments: Section 2.2 should be included in the simulation section.

Response: In Sec2.2, we use an example to demonstrate the effectiveness of the load forecasting model. Sec3.4 is the simulation section, which mainly displays the optimal configuration results. Therefore, we separate the two parts of the content into different section.

6.Reviewer’s comments: In section 3.2.1, there are two objectives, making the problem as a multi-objective optimization problem. Thus, the authors should discuss how to obtain the multi-objective optimization results. The “fmincon” function is not supposed to be the correct solution algorithm.

Response: We overlooked this part in the original manuscript, and we have provided explanations and corrections in Page 9 of the revised version :

Since the model is a multi-objective optimization model and cannot be solved directly, we adopt a two-stage approach. In the first step, we ignore Objective 2 and solve the optimization model with only Objective 1:, keeping the constraints consistent with the original model. We assume that the investment payback period obtained in this step is N years. In the second step, we use Objective 2 of as the single objective function, taking the objective value from the first step as a constraint,i.e., adding the following constraint to the original model's constraints:. The solution obtained in the second step will be the final solution of the model. In the process of solving the optimization model using the above two-step method, each step's optimization was performed using the optimization function 'fmincon' in MATLAB (as shown in Figure 5). The final optimized solution includes the photovoltaic installed capacity, energy storage system capacity and power, power distribution among photovoltaics, energy storage, production plants, and power systems, as well as the business payback period and annual income.

7.Reviewer’s comments:The integration of the forecasting results, especially in future years, into the optimization model should be further clarified.

Response: The power load predicted results have been integrated into the constraints of the optimization model, see Eq.(16). As the answer to question 4, we only need to change the short-term prediction results in the input variables of the optimal configuration model to long-term predictions, then we can obtain the long-term planning results for the enterprises. 

8.Reviewer’s comments:Planning or configuration problem is usually a mix-integral problem, with the number of the equipment as the integers. The authors should discuss why the integer issue is not considered in this paper.

Response: In this paper, we only study the capacity of photovoltaic energy storage and do not investigate the quantity of photovoltaic energy storage. The default quantity is 1. The charging and discharging of energy storage in the model are integer programming problems, depending on the constraints 21-23. Since the energy storage system cannot be operated for charging and discharging at the same time, nor can it be charged with PV and the grid at the same time.

9.Reviewer’s comments:Direct planning or configuration results of the paper is missing.

Response: In Sec3.4, we take a factory as an example to illustrate the results of our model. In the second paragraph of Page10, we provide the optimal configuration of new energy for the factory and also obtain the daily optimal charging and discharging plan for energy storage(see Fig.12). Finally, we obtained the economic benefit analysis results of the enterprise's new energy investment, as shown in Table 1.

10.Reviewer’s comments:The results should be compared with other existing methods to show the innovation.

Response: Before proposing our model, We pointed out the shortcomings of other methods in addressing the main issues in the paper, such as “An optimal configuration model for photovoltaic energy storage is established for users whose transformer capacity is already overloaded or about to overload. Compared with traditional models, the objective function considers the impact of the cost of transformer capacity upgrade and basic electricity price increase.”(At the end of the Introduction section), and “From the historical data, the power load characteristics show a gradual upward trend in a certain period. However, the power load is affected by many factors, such as holidays, seasons and other factors. It has its own unique pattern of change, the traditional time series analysis simulation is difficult to apply.”(At the end of Page 3 ).

11.Reviewer’s comments:The overload issue is not discussed in the case study.

Response: One of the main innovations of this article is to solve the problem of transformer overload by configuring new energy sources. Identify users at risk of overloading through enterprise electricity load forecasting. For these types of enterprises, by configuring new energy, not only can the overload problem be solved, but it also brings great economic benefits, mainly by saving the cost of transformer capacity upgrade and the cost of basic electricity price increase ， given in the objective function, see Eq.(5). Their detailed descriptions can be found below Eq.(6).

12.Reviewer’s comments: The overall English expression should be improved.

Response: We have carefully proof-read the manuscript to minimize typographical, grammatical, and bibliographical errors. Many grammatical or typographical errors have been corrected. 

Response to Reviewer #2

1.Reviewer’s comments: The authors mentioned a proposed overload prediction enterprise methodology for transformers by taking the energy storage system and PV system as examples. Draw and explain your key findings with a load flow diagram and load indication points.

Response: The relationship diagram between the key variables in the optimal configuration model is added in the revised manuscript, see Figure 4.

2.Reviewer’s comments: The authors stated that your non-linear programming models detect the potential consumers who are under overload. How does this reduce the enterprise's basic tariff and electricity prices and improve efficiency? You did not mention the calculations and descriptions.

Response: This article aims to explore customers whose transformer capacity is about to overload through load forecasting, and identify these customers as potential high-quality customers for new energy installation. A new energy optimal configuration model is proposed for these customers. We selected 820 enterprises in a certain region with transformer capacities greater than 300KVA as research objects, and screened out 89 potential users. Under the constraint of not overloading the load after the installation of new energy, we established the optimal configuration model for the enterprises, which can not only solve the overload problem, but also brings great economic benefits, mainly by saving the cost of transformer capacity upgrade and the cost of basic electricity price increase , given in the objective function, see Eq.(5). Their detailed descriptions can be found in Page 8.

3.Reviewer’s comments: If a fault occurs in the systems, how does your methodology work? Is there any protection and identification system on the transformer end?

Response: This system connects energy storage inverters, photovoltaic inverters, and transformer control systems through the Internet of Things to achieve centralized control. By using the optimization model we have established, we integrate photovoltaics, energy storage, and the power grid to meet the electricity needs of enterprises in the optimal way. If a fault occurs in the systems, the transformer will shut down when the enterprise's electrical load exceeds 1.2 times the transformer capacity, otherwise it will continue to operate, which is to protect the transformer.

4.Reviewer’s comments: English grammar corrections are required, particularly on page 19, 1st paragraph. Check the grammar and technical corrections.

Response: We have carefully proof-read the manuscript to minimize typographical, grammatical, and bibliographical errors. Many grammatical or typographical errors have been corrected. 

5.Reviewer’s comments:Apart from the above-mentioned points, the remaining article is technically sound and explained in a good way.

Response: Thank you again for your helpful comments and suggestions.

---

## [Decision Letter · Decision Letter 1]

18 Sep 2024

Identification of transformer overload and new energy planning for enterprises based on load forecasting

PONE-D-24-16376R1

Dear Dr. Han,

We’re pleased to inform you that your manuscript has been judged scientifically suitable for publication and will be formally accepted for publication once it meets all outstanding technical requirements.

Kind regards,

Ch. Rami Reddy, Post Doc

Academic Editor

PLOS ONE

Additional Editor Comments (optional):

Reviewers' comments:

Reviewer's Responses to Questions

**Comments to the Author**

1. If the authors have adequately addressed your comments raised in a previous round of review and you feel that this manuscript is now acceptable for publication, you may indicate that here to bypass the “Comments to the Author” section, enter your conflict of interest statement in the “Confidential to Editor” section, and submit your "Accept" recommendation.

Reviewer #1: All comments have been addressed

Reviewer #2: All comments have been addressed

2. Is the manuscript technically sound, and do the data support the conclusions?

Reviewer #1: Yes

Reviewer #2: Yes

3. Has the statistical analysis been performed appropriately and rigorously? 

Reviewer #1: N/A

Reviewer #2: Yes

4. Have the authors made all data underlying the findings in their manuscript fully available?

Reviewer #1: Yes

Reviewer #2: Yes

5. Is the manuscript presented in an intelligible fashion and written in standard English?

Reviewer #1: Yes

Reviewer #2: Yes

6. Review Comments to the Author

Reviewer #1: The authors have addressed all the concerns and comments from the reviewer. there is no further comment.

Reviewer #2: Reviewed the updated manuscript and pleased to see that you have responded to all the comments and questions raised in the previous round. Your revisions have been well-executed, and the thoroughness with which you addressed the concerns is commendable.

Thank you for your dedication to improving the quality of the work. Your efforts have significantly enhanced the clarity and impact of the manuscript.

7. PLOS authors have the option to publish the peer review history of their article (what does this mean?). If published, this will include your full peer review and any attached files.

Reviewer #1: No

Reviewer #2: No

---

## [Editor Report · Acceptance letter]

23 Sep 2024

PONE-D-24-16376R1 

PLOS ONE

Dear Dr. Han, 

I'm pleased to inform you that your manuscript has been deemed suitable for publication in PLOS ONE. Congratulations! Your manuscript is now being handed over to our production team.

Kind regards, 

on behalf of

Dr. Ch. Rami Reddy 

Academic Editor

PLOS ONE